# Truthful Aggregation of LLMs with an Application to Online Advertising

## Abstract

We study how to aggregate the preferences of multiple agents over LLM-generated replies to user queries. The agents are self-interested and may thus misreport their preferences, and new agents may participate for each new query, making fine-tuning LLMs on their preferences impractical. To address these challenges, we propose an auction mechanism that works without fine-tuning or access to model weights. The mechanism is designed to provably converge to the output of the optimally fine-tuned model as computational resources are increased. The mechanism can also incorporate contextual information about the agents when available, significantly accelerating its convergence. Our mechanism ensures that truthful reporting is the optimal strategy for all agents, while also aligning each agent's utility with her contribution to social welfare – an essential feature for the mechanism's long-term viability. Although our mechanism can be applied in any setting with monetary transfers, our key application is online advertising. In this domain, advertisers try to steer LLM-generated responses based on their interests, while the platform aims to maximize advertiser value and ensure user satisfaction. Experimental results confirm that our mechanism not only converges efficiently to the optimally fine-tuned LLM, but also significantly boosts advertiser value and platform revenue, all with minimal computational overhead.

## 1 Introduction

*Large language models (LLMs)* are becoming ubiquitous – as coding assistants, as chat interfaces, as complements to search engines, and for many other applications [Bommasani et al., 2022]. To ensure their usefulness, it is essential to align LLM outputs closely with user preferences. In general, though, there may be multiple interested parties who disagree over the desired behavior of *the same* language model. How should we guide language model behavior to respect multiple conflicting preferences?

As a practical motivation for our work, we focus on online advertising. Over the years, advertising has established itself as the main source of revenue for large tech companies such as Google, Meta, and Twitter. In 2023, Meta's advertising revenue of 132 billion USD was over 97% of its total revenue, with auctions being the workhorse mechanism determining the placement and prices of commercial content [Varian, 2007, Edelman et al., 2007]. As existing platforms begin to serve more LLM-generated content, new auction mechanisms have to be created for this application.

In this paper, we present a new auction mechanism for this problem. The agents are paying not for some particular item or bundle, but rather to influence the output generated by an LLM in a direction closer to their own preferences. While our mechanism could be of interest whenever one has to aggregate the preferences of multiple self-interested agents over LLM behavior (as long as it is reasonable to charge monetary payments), we see online advertising as the most salient setting of interest. For this reason, we use the terms *advertiser* and *agent* interchangeably.

Submitted to 38th Conference on Neural Information Processing Systems (NeurIPS 2024). Do not distribute.

## 1.1 Concrete problem setting

Consider a situation where a user has queried an LLM for a specific task. We assume the following:

- There is a *reference LLM* that can produce useful replies to the user's query.
- There are *agents* who would like to be represented in the reply back to the user.

In our model, the agents can be represented either via their own LLMs or directly with a *reward function*, similar to the function used to fine-tune their LLMs in the *Reinforcement Learning from Human Feedback* pipeline [Azar et al., 2023]. For this reason, we refer to an agent's value for a reply as her *reward*. The auctioneer's task is, given the user's query and the agents' preferences, to *produce a reply that is useful for the user, while at the same time generating high rewards for the agents*.

Our running example will be the following. A user is interested in baking and queries an LLM "How do I bake cookies." There are two interested agents, each a different advertiser, who would like to influence the response to the user: *EasyBake*, a company that produces baking ingredients and *KitchenFix*, a company that produces kitchen equipment.

## 1.2 Challenges

**Good outcomes:** Our mechanism must produce useful outcomes, in the sense that agents receive high rewards, but without steering the LLM's behavior too far from that of the "reference" LLM that produces useful replies for the user. We formalize this trade-off in Section 3.

**Technical feasibility and practicality:** Duetting et al. [2023] argued that auction solutions must be compatible with existing LLM technology, using only "information obtainable from current models" in such a way that they are "easy to integrate into the system" and relying only on "easy manipulations of the LLM outputs." We strongly agree with these desiderata and adopt them for our work. A related point is that it must be computationally feasible to run the auction mechanism repeatedly and with many different agent preferences. In particular, we cannot afford the expensive process of fine-tuning an LLM's weights for each possible user query. We explain how our mechanism satisfies these requirements in Section 4.1. The key idea is that our mechanism works only by post-processing multiple LLM outputs – it requires only "API access" and does not require modifying or even viewing the model weights.

**Mechanism design desiderata:** Agents might have an interest in misreporting their preferences to get a better outcome, analogous to over-bidding or under-bidding in traditional auctions. We design an auction where there is no such incentive for any agent, no matter what the other agents do: a mechanism with this property is called *strategyproof*. In traditional auction settings, another sensible property is *ex-post individual rationality (IR)*, where participating and reporting truthfully guarantees that an agent is at least as well off as not participating. In Section 5.2, we discuss unique properties of our setting and explain why ex-post IR is unattainable in our setting. In Section 5.3.1 we show that our mechanism is, however, "almost IR" and in Section 6.2.1, we experimentally show that our mechanism is *ex-ante* IR, meaning that an agent is in expectation better off by participating.

## 1.3 Overview of Contributions

We present a novel auction mechanism designed to aggregate the preferences of multiple self-interested agents over LLM-generated replies. We make several key contributions. First, our mechanism allows for an interpretable, principled way of balancing between the usefulness of the produced reply to the user and the agents' preferences (Section 3.1). Second, it is the only mechanism in the literature that converges to the optimal distribution as computational resources are increased (Corollary 4.1). Third, it can incorporate contextual information, similar to how sponsored search auctions utilize advertiser descriptions. This boosts performance (Section 6.2.1), increasing value for the agents and accelerating convergence.

Fourth, our mechanism is strategyproof, in the sense that it is an optimal strategy for each agent to truthfully report her preferences (Theorem 5.1). Crucially, this is true even if the allocation rule has not converged to optimality. Fifth, our mechanism is equitable in the sense that each agent's utility is proportional to her contribution to the social welfare. This alignment is vital for the long-term success of a market for this setting, incentivizing the most relevant agents to participate in the auction.

The flagship application of our mechanism is in online advertising. Our experiments in this domain (Section 6) demonstrate that our mechanism converges to the optimal distribution with low computational cost, generating significant value for the advertisers and revenue for the auctioneer. Moreover, it maintains the aforementioned equity property while ensuring positive utility for advertisers.

# 2 Prior Work

Duetting et al. [2023] were the first to suggest an auction mechanism for LLMs. The authors proposed a sequential mechanism, where the output sequence is generated on a token-by-token basis and the advertisers bid each time for their LLM to generate the next token. However, their approach suffers from significant limitations: (i) For a given prompt, an advertiser's spend grows with the length of the generated sequence. (ii) Advertisers suffer from the *exposure problem*: Adding a "not" to a sequence completely changes its meaning, and an advertiser could have paid a significant amount for the sequence generated up to some point, not expecting a negation in its continuation. (iii) The mechanism is easily manipulable if the assumption that advertisers cannot misreport their LLMs is dropped. (iv) The authors prove that an advertiser bidding higher leads to an aggregate distribution for the next token that she prefers; however, they do not provide any guarantees on the distribution of the resulting full output sequence. Our mechanism handles all of these limitations.

Dubey et al. [2024] proposed a generalization of the position auction [Varian, 2007, Edelman et al., 2007] to a setting where each advertiser is interested in having their text ad displayed, and an LLM module coupled with an auction module work together to merge the ads into a single summary in an incentive-compatible way. Their mechanism takes as input the ad creative of each advertiser and, given a prediction model of click through rates, creates a summary of those creatives that maximizes advertiser welfare, defined as the dot product of the advertisers' values per click times their predicted click through rate. By comparison, our mechanism takes as input directly the advertisers' rewards for some sequences (or equivalently, the probabilities of those sequences with respect to the advertiser LLMs, see Section 4.1), and outputs a reply that follows in the limit the theoretically optimal distribution, maximizing expected advertiser reward subject to remaining close to the distribution induced by a reference LLM responsible for creating a useful reply for the user.

Feizi et al. [2024] presented an abstract design for an LLM advertising system and detailed a number of research challenges that would have to be overcome in the course of implementation. Conitzer et al. [2024] drew connections between *computational social choice* and LLM alignment. Social choice theory is closely related to auction design, with different emphases: it is typical in social choice to think in terms of ordinal rather than cardinal preferences, and monetary payments are typically not charged. Fish et al. [2023] presented work in the opposite direction: how can LLMs be used to solve problems in social choice? Harris et al. [2024] studied Bayesian persuasion in an abstract setting where a "simulator" (for example, a realistic LLM) of the agent is available.

# 3 Framing Sequence Generation as a Mechanism Design Problem

## 3.1 Formal Model

A *user* issues a query $x$. There is a *reference LLM* $\pi_{\text{ref}}$ that the auctioneer aims not to deviate from too much (e.g., because it is responsible for providing useful replies to the user). Additionally, there is a set $N$ of $n$ agents (e.g., advertisers) who have their own preferences for the reply (i.e., a token sequence) that will be returned to the user. We use the terms *sequence* and *reply* interchangeably.

An LLM can be abstracted as a mapping from (partial) token sequences to a distribution over the next token, or equivalently as an implicit probability distribution over token sequences. We use this second abstraction, i.e., $\pi_i(y|x)$ denotes the probability that agent $i$'s LLM $\pi_i$ assigns to output sequence (i.e., reply) $y$ for the user query $x$.

We let $r_i(x, y)$ denote agent $i$'s *reward* for sequence $y$, given query $x$.[1] Informally, the auctioneer's goal is to sample the final sequence from a distribution that optimizes the agents' expected rewards

---

[1] In theory, the agent's reward for a generated sequence could also depend on user-specific information, but we abstract that away. Equivalently, we assume that the reward function $r_i$ provided by agent $i$ is specific to the given user that asked the query $x$.

without substantially diverging from $\pi_{\text{ref}}$. Formally, the goal is to maximize:

$$J(\pi) = \mathbb{E}_{y \sim \pi} \left[ \sum_{i \in N} r_i(x, y) \right] - \tau D_{\text{KL}}(\pi(\cdot|x) || \pi_{\text{ref}}(\cdot|x)) \tag{1}$$

where $\tau > 0$ is a hyperparameter enabling the auctioneer to control the trade-off between producing replies more faithful to the reference policy or with higher reward for the agents, and $D_{\text{KL}}$ refers to the Kullback-Leibler divergence.

This objective mirrors the standard Reinforcement Learning from Human Feedback (RLHF) approach [Ziegler et al., 2020], but replaces the human feedback reward function $r_{\text{HF}}(x, y)$ with the sum of the agents' rewards. For an overview of RLHF, we recommend Rafailov et al. [2023, §3].

It is established [Rafailov et al., 2023] that the optimal solution to the optimization problem in (1) is:

$$\pi_r(y|x) = \frac{1}{Z(x)} \pi_{\text{ref}}(y|x) \exp \left( \frac{1}{\tau} \sum_{i \in N} r_i(x, y) \right), \tag{2}$$

where $Z(x) = \sum_{y \in T^*} \pi_{\text{ref}}(y|x) \exp \left( \frac{1}{\tau} \sum_{i \in N} r_i(x, y) \right)$ is the partition function ensuring that $\pi_r(\cdot|x)$ is properly normalized.

Let $\vec{R}$ be the set of all possible reports by the agents. A *mechanism* is defined as a pair $(\pi, p)$. The *allocation rule* $\pi : \vec{R} \to \Delta(T^*)$ maps any report profile $\vec{r} = (r_1, r_2, \ldots, r_n) \in \vec{R}$ of the agents' rewards to a distribution over sequences $\delta(T^*)$. We denote the agents' *aggregate reward* as $r(x, y) = \sum_{i=1}^{n} r_i(x, y)$, and their reward profile as $\vec{r}(x, y) = (r_1(x, y), r_2(x, y), \ldots, r_n(x, y))$. The *payment rule* $p : \vec{R} \to \mathbb{R}^n$ maps any report profile of the agents' rewards to a payment profile $\vec{p}$, where $\vec{p}_i$ is the payment of the $i$-th agent to the mechanism.

A mechanism is *strategyproof* if and only if, for any agent $i \in N$, reporting her rewards truthfully is always optimal for her, no matter the other agents' reports. More formally:

**Definition 3.1** (Strategyproof Mechanism). *A mechanism $(\pi, p)$ is dominant strategy incentive compatible or strategyproof iff for all agents $i \in N$, for all true rewards $\vec{r}_i \in \vec{R}_i$, for all reports $\vec{r}_{-i} \in \vec{R}_{-i}$ by the other agents and for all possible agent $i$ misreports $\vec{r}_i' \in \vec{R}_i$: $\mathbb{E}_{y \sim \pi(\vec{r})}[u_i(y, \vec{r}_i, \vec{r}_{-i}; r_i, x)] \geq \mathbb{E}_{y' \sim \pi(\vec{r}_i', r_{-i}, x)}[u_i(y, \vec{r}_i', \vec{r}_{-i}; \vec{r}_i)]$, where $u_i(y, \vec{r}_i, \vec{r}_{-i}; r_i, x) = r_i(x, \vec{y}) - p_i(\vec{r})$ is the utility of agent $i$ for sequence $y$ to be returned in the report profile $\vec{r} = (\vec{r}_i, \vec{r}_{-i})$ when the agent's reward is $r_i(x, \cdot)$.*

## 3.2 Why not use VCG?

The most celebrated auction mechanism is the Vickrey-Clarke-Groves (VCG) mechanism [Vickrey, 1961, Clarke, 1971, Groves, 1973]. VCG's allocation rule selects the outcome that maximizes the sum of all agents' values.[2] The VCG mechanism has a corresponding payment rule to incentivize truthful reporting: it charges each agent the total reduction in value for the other agents that her participation in the mechanism caused. One could choose the single sequence maximizing the regularized reward in Equation (1) and then charge VCG payments to get a truthful mechanism. Alternatively, one could apply VCG in the distribution space. In that case, the allocation would be the distribution in Equation (2), and each agent's payment would be her expected externality. With either choice, the VCG mechanism would be strategyproof and select the optimal outcome.

In our setting however, VCG is not a viable option: VCG's allocation rule requires calculating the *exact* optimal solution to the optimization problem, which is hopeless for choosing an LLM to maximize Equation (1) and even difficult for choosing a single optimal sequence. If a suboptimal solution is chosen, VCG's strategyproofness is no longer guaranteed [Nisan and Ronen, 2007, 1999, Lehmann et al., 2002]. Thus, VCG is not a suitable mechanism in our domain.

# 4 Our Mechanism: Allocation Rule

In this section, we introduce our mechanism's allocation rule. The high level idea is that first, a set of $M$ *candidate sequences* are generated based on some LLM responsible for that task $\pi_{\text{gen}}$. Then, the

---

[2]Note that in Equation (1) the regularization term can also be interpreted as an agent, with value for sequence $y$ of $-\tau \pi(y|x) \log \frac{\pi(y|x)}{\pi_{\text{ref}}(y|x)}$. In the rest of the paper, the term agents will refer only to the advertisers.

---

**Algorithm 1:** Context-Aware Allocation Rule

---

**Input:** User prompt $x$, reference LLM $\pi_{\text{ref}}$, context-aware LLM used for candidate sequence generation $\pi_{\text{gen}}$, advertiser reward functions $\vec{r}$, advertiser descriptions $\vec{c}$, sentences to sample $M$, regularization parameter $\tau$

**Output:** Output sequence $y$ drawn according to the optimal distribution as defined in Equation (1) for the aggregate reward function $r(x, y) = \sum_{i=1}^{N} r_i(x, y)$

---

**1** Sample $y_j \sim \pi_{\text{gen}}(\cdot|x; \vec{c})$, $1 \leq j \leq M$

**2** Calculate $r(x, y_j) = \sum_{i=1}^{N} r_i(x, y_j)$, $1 \leq j \leq M$

**3** Sample $y \sim \text{softmax}\left(\frac{r(x,y_1)}{\tau} + \log \frac{\pi_{\text{ref}}(y_1|x)}{\pi_{\text{gen}}(y_1|x;\vec{c})}, \ldots, \frac{r(x,y_M)}{\tau} + \log \frac{\pi_{\text{ref}}(y_M|x)}{\pi_{\text{gen}}(y_M|x;\vec{c})}\right)$

**4** **return** *Output sequence $y$*

---

probability of returning each candidate sequence is re-weighted based on the advertisers' reports and the reference LLM $\pi_{\text{ref}}$ so that in the limit as $M \to \infty$, the probability of returning each sequence converges to its probability under the optimal distribution of Equation (2). This approach shares similarities with the rejection sampling approach which has been used at various points of the LLM training pipeline, e.g., Bai et al. [2022], Touvron et al. [2023]. We defer all proofs to Appendix A.

**Corollary 4.1.** *The limiting policy $\pi_{\theta,M}(\cdot|x)$ induced by Algorithm 1 is the KL regularized policy that optimizes the aggregate reward function:*

$$\lim_{M \to \infty} \pi_{\theta,M}(\cdot|x) = \arg\max_{\pi} \mathbb{E}_{y \sim \pi(\cdot|x)}[r(x, y)] - \tau D_{KL}(\pi||\pi_{ref}) \tag{3}$$

Based on Corollary 4.1, our allocation rule converges to the optimal distribution for *any* LLM $\pi_{\text{gen}}$, as long as it generates any sequence that has a non-zero probability under the optimal distribution in the limit. The obvious choice is $\pi_{\text{ref}}$. However, the practical convergence rate for that choice would be very slow: for computationally feasible values of $M$, it's improbable for $\pi_{\text{ref}}$ to generate sequences with high rewards for the advertisers. Consider our running example where the user prompt is "How do I bake cookies?" and the advertisers in question are "EasyBake" and "KitchenFix." The advertisers have higher rewards for sequences that explicitly mention their brands. But, for computationally feasible values of $M$, if we used $\pi_{\text{ref}}$ to generate the candidate sequences, Algorithm 1 would sample the final sequence from a set of candidate sequences which would likely not mention their brands.

To address this and related challenges, we generate candidate sequences not from $\pi_{\text{ref}}(\cdot|x)$, but instead from a *context-aware LLM*, $\pi_{\text{gen}}(\cdot|x; \vec{c})$. The instance-specific context $\vec{c}$ is designed to bridge the gap between the parts of the output space favored by $\pi_{\text{ref}}$ and those valued by the advertisers.

In our application of integrating advertisers' interests into LLM outputs, $\vec{c}_i$ is a context-specific description of the $i$-th advertiser. These descriptions, supplied by the advertisers themselves, should be easily verifiable and factually accurate, akin to "EasyBake: producing baking ingredients", or "KitchenFix: currently offering promotions on baking ovens."[3] This approach is analogous to search engine optimization in sponsored search advertising, where advertisers supply and potentially optimize their own descriptions to influence how they are presented by the auction mechanism.

Our experiments in Section 6.2.1 demonstrate that, within computationally reasonable limits, using the context-aware LLM to generate candidate sequences achieves substantially higher rewards and utility for the advertisers, increased revenue for the auctioneer, and faster convergence. In the rest of the paper, we refer to using the reference and context-aware LLMs as the baseline and context-aware versions of our mechanisms, respectively.

## 4.1 Applicability and Practical Considerations of the Mechanism

**Input Methods and Computational Efficiency**    Both our allocation and payment rule (introduced in Section 5) do not depend on the agents' full reward functions, but only on their rewards for the candidate sequences. Thus, our mechanism requires only "API access" to the involved LLMs without fine-tuning or access to their weights. In Appendix B.3, we establish a mapping between an

---

[3]A practical way of implementing $\pi_{\text{gen}}(\cdot|x; c)$ given $\pi_{\text{ref}}(\cdot|x)$ is to augment the input $x$ to the reference LLM with the advertiser descriptions. In our example: "Try to mention ⟨advertiser x⟩, ⟨advertiser x description⟩.'

agent's LLM and her implicit reward function, allowing our mechanism to use as inputs sequence probabilities (i.e., LLM inference calls) instead of rewards.

Our mechanism can be integrated with computationally efficient methods for eliciting agents' reward functions and sampling from the base LLM, reducing computational overhead. For instance, Li et al. [2024] demonstrate that a simple linear function on a model's embedding space can approximate a model's reward function. If agents' reward functions are represented this way, an auction instance with $M$ candidate sequences and $n$ agents would require only $n$ LLM inference calls instead of $n \cdot M$, and $N \cdot m$ linear multiplications, reducing overhead by a factor of $n$.

**Parallelization** The generation and evaluation of each candidate sequence are independent processes. This independence allows our mechanism to be fully parallelized, ensuring that the response time for a user query is comparable to that of querying an LLM directly.

# 5 Our Mechanism: Payment Rule

In this section, we first show how the allocation rule from Section 4 can be combined with an appropriate payment rule so that the resulting mechanism is incentive compatible (Section 5.1). Then, we detail how auctions for LLM-generated content differ from standard auctions (Section 5.2). Taking those differences into account, we create a payment offset, so that the mechanism is more equitable while maintaining its incentive compatibility (Section 5.3). We defer all proofs to Appendix B.

## 5.1 Incentive Compatible Payments through Convexity

**Theorem 5.1.** *Let $\vec{r}_{-i}$ be the reward profile of all agents other than $i$, and $\vec{\beta}_{-i}$ the aggregate reports of all other agents and the reference and context-aware LLMs. Then, the allocation rule induced by Algorithm 1 can be combined with a payment rule $p : \vec{R} \to \mathbb{R}^n$ such that in the mechanism $(\pi, p)$ for any agent $i \in N$, report profile $\vec{\beta}_{-i}$ and set of generated candidate sequences, reporting truthfully maximizes agent $i$'s expected utility, with the expectation taken over the draw of the final sequence from the set of candidate sequences. Agent $i$'s utility in $(\pi, p)$ under truthful bidding is:*

$$U_{C,i}(\vec{r}_i; \vec{\beta}_{-i}) = \tau \log \left( \sum_{j=1}^{M} \exp \left( \frac{1}{\tau} \sum_{k \in N} r_k(x, y_j) \right) + \log \frac{\pi_{ref}(y_j|x)}{\pi_{gen}(y_j|x; \vec{c})} \right) + C, \ C \in \mathbb{R} \quad (4)$$

Note that, based on Theorem 5.1, in our mechanism it is *always* a dominant strategy for an agent to report truthfully. Crucially, this is not the case for VCG, where truthful reporting would be optimal only if the allocation rule had converged to the optimal distribution as defined in Equation (2).

## 5.2 Differences from Standard Auction Settings

Standard auction environments rely on assumptions that do not apply to auctions for LLM-generated content. Key differences include: (i) *Non-Negative Values:* Standard auctions assume agents' values are non-negative due to having zero value for the empty bundle and free disposal. In our setting, an agent's reward can go negative based on the discrepancy between her LLM and the reference LLM. (ii) *Agent-Specific Allocations:* Standard auctions allocate different bundles to different agents. In our setting, a single sequence is produced, and agents' rewards depend on its probability with respect to their LLMs. (iii) *Zero Utility for Non-Participation:* In standard auctions, not participating yields zero utility. Here, non-participation can result in negative utility since the produced sequence may be unfavorable to the non-participating agents. For more details, see Appendix B.3.

## 5.3 Agent-Specific Utility Offset

In this section, we introduce our mechanism's utility (and thus payment) offset. Our goal is to maintain the nice properties of our mechanism, namely incentive compatibility and convergence to the optimal distribution, while also achieving two additional properties that we argue are important for the long-term success of a market for LLM aggregation. That offset is $C = -\tau \log \left( \sum_{j=1}^{M} \exp \left( \vec{\beta}_{-i,j} \right) \right)$.

This offset has an intuitive explanation: charge each agent her utility for bidding zero for all candidate sequences according to Equation (4). This maintains strategyproofness, while ensuring the following properties:

- *"Almost IR:"* An agent with weakly positive reward for all generated candidate sequences, has weakly positive expected utility, for all reports by the other agents. In particular, an agent who has zero utility for all outputs is guaranteed zero utility from the mechanism.
- *"What you give is what you get:"* The ex-interim expected utility of an agent is monotone in how well-aligned her exponentiated reward for the sequences is with the interim allocation rule if she were to not participate.

### 5.3.1 Our Mechanism is "Almost Individually Rational"

In Appendix B.2 we explain why the standard notion of individual rationality (i.e., weakly positive utility from participation in the mechanism) encountered in most auction settings is impossible to achieve in this domain while converging to the optimal distribution and maintaining strategyproofness. Then, we explain how, with our payment offset, our mechanism is "almost IR:" In Lemma B.1 we prove that the ex-interim utility of an agent who has zero reward for all candidate sequences and bids truthfully is deterministically zero, i.e., agents that do not contribute to the social welfare (but also do not detract from it) have zero utility. Similarly, in Lemma B.2 we prove that if an agent's reward for all candidate sequences is (weakly) positive, then her ex-interim utility is (weakly) positive.

**Remark 1.** *In Section 6 we will experimentally show that this offset, coupled with our generation of candidate sequences based on the context-aware LLM, results in both high expected rewards and positive expected utility for the agents for participating in the mechanism, i.e., ex-ante individual rationality, while at the same time yielding significant revenue to the auctioneer.*

### 5.3.2 "What you give is what you get"

Our choice of allocation rule (which is the only allocation rule over a finite set of sequences that converges to the optimal distribution), combined with the fact that the allocation rule is the gradient of the utility to ensure truthfulness, means that agent utilities must also be the same up to potentially agent-specific offsets as indicated by Equation (4).

However, not all agents contribute equally to the social welfare of the final outcome. In appendix B.4 we detail why implementing the mechanism without a carefully-designed offset would lead to a kind of "reverse market unraveling:" as long as an agent's utility in Equation (4) is positive, it would incentivize agents for whom the user query is unrelated to participate in the auction. This would reduce everyone's expected utility, and thus disincentivize the relevant agents to participate, leading to sequences with worse expected rewards for the agents and usefulness for the user. Thus, it is crucial to align each agent's utility with her contribution to the social welfare.

**Lemma 5.2.** *For the offset $C = -\tau \log \left( \sum_{j=1}^{M} \exp \left( \vec{\beta}_{-i,j} \right) \right)$ agent $i$'s ex-interim utility is:*

$$U_i(\vec{r}_i; \vec{\beta}_{-i}) = \tau \log \left( \sum_{j=1}^{M} \exp \left( \frac{r_i(x, y_j)}{\tau} \right) \pi_{int}(y_j | x; \vec{r}_{-i})_j \right) \tag{5}$$

In words, Lemma 5.2 proves that every agent's ex-interim utility is monotonic in how well-aligned the interim allocation rule $\pi_{\text{int}}$ (i.e., the probability of returning each of the already generated candidate sequences) without her and her exponentiated rewards for the candidate sequences are.

**Remark 2.** *In Appendix C.4 we will experimentally show that, with the offset in Section 5.3, there is a strong positive correlation between an advertiser's contribution to the social welfare and her expected utility gain from participation in the mechanism, and that the relationship between the two quantities is quite linear. Additionally, we will show that the resulting mechanism is ex-ante IR.*

## 6 Experiments

In this section, we experimentally evaluate the performance of our proposed mechanism. The experiment focuses on the flagship application of our mechanism, integrating advertisers' interests into LLM-generated replies to user queries while ensuring that the replies are useful.

## 6.1 Experiment Setup

We create synthetic instances consisting of user queries (e.g., "How do I bake cookies?") and advertisers (e.g., "KitchenFix, producing kitchen appliances"). We use Llama-2-7b-chat-hf as the reference LLM [Touvron et al., 2023]. We create advertisers' LLMs by adding advertising instructions to the reference LLM. The context-aware LLM is created as described in Footnote 3.

Following Rafailov et al. [2023], advertisers' reward functions are defined as $r_i(x, y) = \log \frac{\pi_i(y|x)}{\pi_{\text{ref}}(y|x)}$. For the auctioneer's objective, we set $\tau = 1$ in Equation (1), balancing advertisers' rewards and sequence divergence from the reference LLM.

We use 50 user queries, each with two advertisers, and test each query on 25 different random seeds, resulting in 1250 instances. Following Li et al. [2024], Rozière et al. [2024] we sample from all LLMs using temperature 0.8 and top-p 0.95. For full experimental details, see Appendix C.1.

## 6.2 Experimental Results

### 6.2.1 Evaluating the Effectiveness of Incorporating Context into the Mechanism

To illustrate how the context-aware mechanism enhances the relevance of responses for advertisers, we compare outputs from both mechanisms in Appendix C.3. Notably, only the context-aware mechanism successfully incorporates advertisers into the replies.

Our main results are illustrated in Figure 1. In Figure 1a, we plot the log probability (with respect to the optimal distribution in Equation (2)) of the sequences returned by our mechanism as a function of the number of candidate sequences generated. We evaluate the effectiveness of incorporating contextual information by comparing the context-aware version of our mechanism to the baseline version that uses the reference LLM to generate candidate sequences. To benchmark our results, we estimate the log probability of sampling sequences from the theoretically optimal distribution.[4]

In Figure 1b, we plot the total advertiser *reward gain from participation* in the mechanism. We define the reward gain for an advertiser as the difference in her expected reward (with the expectation taken over the sampling of the returned sequence from the candidates) when she is participating in the mechanism, versus when she is not, i.e. $r_{i,\text{gain}}(x) = \sum_{j \in M} r_i(x, y_j) \cdot \pi_{\text{int}}(y_j|x) - \sum_{j \in M} r_i(x, y_{-i,j}) \cdot \pi_{\text{int}}(y_{-i,j}|x)$, where $y_{-i} = (y_{-i,1}, \ldots, y_{-1,M})$ represents the candidate sequences that would have been generated had agent $i$ not participated.[5] Again, we compare the outcomes using the context-aware versus the baseline mechanism, reinforcing the added value of integrating context.

In Figure 1a we observe that both for the context-aware and baseline versions of our mechanism, the log probability of the returned sequence increases with the number of candidate sequences generated. This result is in line with our theoretical analysis in Section 4, where we proved that both versions of the mechanism converge to the optimal distribution in the limit. At the same time, we observe that incorporating context into the mechanism is significantly more efficient. Notably, our context-aware mechanism can achieve higher log probability with respect to the optimal distribution using only four candidate sequences than the baseline version can achieve with 20. Additionally, with only 20 generated candidate sequences, our context-aware mechanism is able to almost match our estimate of the log probability of sampling from the theoretically optimal distribution.

At the same time, Figure 1b demonstrates that our context-aware mechanism significantly increases advertiser reward, with the benefits scaling rapidly with the number of sequences generated. In contrast, the baseline version of our mechanism is unable to increase advertiser reward within a computationally feasible number of generated sequences. To conclude, our mechanism's support of

---

[4]Note that the closed form solution of Equation (13) allows as to evaluate the probability of sentences with respect to the optimal solution, but it does not enable us to sample from that distribution. Sampling from that distribution would require using reinforcement learning to train the optimal LLM on the agents' aggregate reward function, which is computationally infeasible for the number of problem instances that we test. So instead, we generate replies from the reference LLM, and evaluate them based on the induced probabilities of the reference LLM, for which the reference LLM is the optimal one. This serves as a proxy for the log probabilities that we should expect if we were to draw replies from the optimally fine-tuned model for each query.

[5]To reduce computational costs, we estimate an advertiser's reward for not participating based on her expected reward over the already generated sequences in which her brand is not mentioned by name (motivated by the fact that, if she does not participate, her brand will not be mentioned by name).

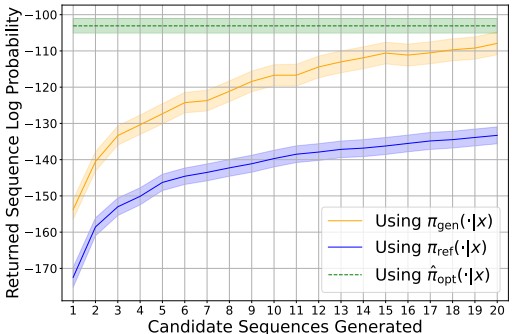
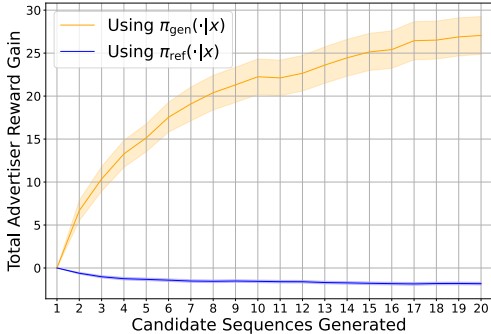

(a) Log probability of the returned sequence as a function of the number of candidate sequences, comparing against the reference LLM and a proxy for the optimal distribution ($\hat{\pi}_{\text{opt}}$).

(b) Total advertiser reward gain from participation as a function of the number of candidate sequences.

Figure 1: Returned sequence log probability and total advertiser reward gain from participation as a function of the number of candidate sequences generated using both, $\pi_{\text{ref}}$ and $\pi_{\text{gen}}$, and $\hat{\pi}_{\text{opt}}$, a proxy of the optimal distribution. Averaged over 1250 runs including 95% CIs.

context-aware LLMs enables it to quickly converge to the theoretically optimal distribution, while at the same time generating significant rewards for the advertisers.

In Appendix C.4, we investigate the effectiveness of our payment rule and offset from Section 5.3. Our experiments demonstrate that our payment rule with the offset makes the mechanism ex-ante IR in practice, i.e., the expected reward gain from participation in the mechanism is positive (Figure 2), and captures a significant portion of the value generated for the advertisers as revenue, which increases rapidly with the number of generated candidate sequences (Figure 2). Additionally, it aligns each advertiser's utility with her contribution to the social welfare (Figure 3). In Appendix C.5 we show that the introduction of the offset makes the relationship between an advertiser's reward and utility gain from participation significantly more linear and positively correlated.

## 7 Conclusion

We have introduced a novel auction mechanism for aggregating preferences over LLM outputs, which provably converges to the theoretically optimal distribution. It also facilitates a principled and interpretable method for balancing participants' expected rewards with the divergence from a reference policy. Thus, our mechanism is particularly well-suited for online advertising, allowing the integration of advertiser LLMs with a reference LLM responsible for generating user-centric replies.

Our carefully engineered payment rule removes any incentive to exaggerate or misreport preferences, achieving the central mechanism design goal of incentive compatibility. While ex-post individual rationality is incompatible with incentive compatibility in this context, we experimentally show that our mechanism is ex-ante individually rational and "almost individually rational" in a certain sense. Furthermore, it ensures that each agent's utility gain is proportionate to her contribution to social welfare, an alignment we argue is important for the long-term success of a mechanism in this setting.

Experimentally, we have demonstrated that by incorporating contextual information, our mechanism's outputs rapidly converge to the optimal distribution, generating significant value for the participants while also effectively recapturing a considerable portion of this value as revenue for the auctioneer. These findings demonstrate the practical efficacy and potential of our approach in real-world settings.

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

# A  Proofs and Theorems from Section 4

In this section, we present all omitted theorems and proofs from Section 4.

**Theorem A.1.** *Let $\pi_{\theta,M}(y|x)$ be the probability of sampling output sequence $y$ for input sequence $x$ according to Algorithm 1, where $\theta$ is the vector of all input parameters and $M$ is the number of candidate sequences generated. Given the agents' reports $\vec{r} \in \vec{R}$, the policy induced by the mechanism approaches the following limit:*

$$\lim_{M \to \infty} \pi_{\theta,M}(y|x) = \pi_{ref}(y|x) \frac{\exp(r(x,y)/\tau)}{\mathbb{E}_{y' \sim \pi_{ref}(\cdot|x)}[\exp(r(x,y')/\tau)]} \tag{6}$$

*Theorem A.1 Proof.* Let $\pi_{\theta,M}(y|x, \{y_j\}_{j=1}^M)$ be the probability of returning output sequence $y$ for input sequence $x$ according to Algorithm 1 conditioned on the set of generated candidate sequences being $\{y_j\}_{j=1}^M$. Additionally, let $\pi_{\text{gen}}(\{y_j\}_{j=1}^M | x; \vec{c})$ be the probability of the context-aware model $\pi_{\text{gen}}$ generating the candidate sequences $\{y_j\}_{j=1}^M$, given the context $\vec{c}$ and the user query $x$.

First, note that we can write the density of $\pi_{\theta,M}$ as follows:

$$\pi_{\theta,M}(y|x) = \sum_{\{y_j\}_{j=1}^M \in Y^M} \pi_{\theta,M}(y|x, \{y_j\}_{j=1}^M) \pi_{\text{gen}}(\{y_j\}_{j=1}^M | x; \vec{c})$$

$$= \mathbb{E}_{\{y_j\}_{j=1}^M \sim \pi_{\text{gen}}(\cdot|x;\vec{c})} \left[ \pi_{\theta,M}(y|x, \{y_j\}_{j=1}^M) \right]$$

$$= \mathbb{E}_{\{y_j\}_{j=1}^M \sim \pi_{\text{gen}}(\cdot|x;\vec{c})} \left[ \sum_j \mathbb{I}\{y_j = y\} \frac{\exp\left(\frac{r(x,y_j)}{\tau} + \log \frac{\pi_{\text{ref}}(y_j|x)}{\pi_{\text{gen}}(y_j|x;\vec{c})}\right)}{\sum_{\zeta \in \{y_j\}_{j=1}^M} \exp\left(\frac{r(x,\zeta)}{\tau} + \log \frac{\pi_{\text{ref}}(\zeta|x)}{\pi_{\text{gen}}(\zeta|x;\vec{c})}\right)} \right]$$

$$= \mathbb{E}_{\{y_j\}_{j=1}^M \sim \pi_{\text{gen}}(\cdot|x;\vec{c})} \left[ \sum_j \mathbb{I}\{y_j = y\} \frac{\frac{\pi_{\text{ref}}(y_j|x)}{\pi_{\text{gen}}(y_j|x;\vec{c})} \exp\left(\frac{r(x,y_j)}{\tau}\right)}{\sum_{\zeta \in \{y_j\}_{j=1}^M} \frac{\pi_{\text{ref}}(\zeta|x)}{\pi_{\text{gen}}(\zeta|x;\vec{c})} \exp\left(\frac{r(x,\zeta)}{\tau}\right)} \right]$$

$$= \mathbb{E}_{\{y_j\}_{j=1}^M \sim \pi_{\text{gen}}(\cdot|x;\vec{c})} \left[ \frac{\sum_j \mathbb{I}\{y_j = y\}}{\sum_{\zeta \in \{y_j\}_{j=1}^M} \frac{\pi_{\text{ref}}(\zeta|x)}{\pi_{\text{gen}}(\zeta|x;\vec{c})} \exp\left(\frac{r(x,\zeta)}{\tau}\right)} \right] \frac{\pi_{\text{ref}}(y|x)}{\pi_{\text{gen}}(y|x;\vec{c})} \exp\left(\frac{r(x,y)}{\tau}\right)$$

$$= \mathbb{E}_{\{y_j\}_{j=1}^M \sim \pi_{\text{gen}}(\cdot|x;\vec{c})} \left[ \frac{\frac{1}{M}\sum_j \mathbb{I}\{y_j = y\}}{\frac{1}{M}\sum_{\zeta \in \{y_j\}_{j=1}^M} \frac{\pi_{\text{ref}}(\zeta|x)}{\pi_{\text{gen}}(\zeta|x;\vec{c})} \exp\left(\frac{r(x,\zeta)}{\tau}\right)} \right] \frac{\pi_{\text{ref}}(y|x)}{\pi_{\text{gen}}(y|x;\vec{c})} \exp\left(\frac{r(x,y)}{\tau}\right)$$

Taking the limit as $M \to \infty$ and using the Law of Large Numbers (the sequences are i.i.d.):

$$\lim_{M \to \infty} \pi_{\theta,M}(y|x) = \lim_{M \to \infty} \mathbb{E}_{\{y_j\}_{j=1}^M \sim \pi_{\text{gen}}(\cdot|x;\vec{c})} \left[ \frac{\pi_{\text{gen}}(y|x;\vec{c})}{\mathbb{E}_{\zeta \sim \pi_{\text{gen}}(\cdot|x)}\left[\exp\left(\frac{r(x,\zeta)}{\tau}\right) \frac{\pi_{\text{ref}}(\zeta|x)}{\pi_{\text{gen}}(\zeta|x;\vec{c})}\right]} \right] \frac{\pi_{\text{ref}}(y|x)}{\pi_{\text{gen}}(y|x;\vec{c})} \exp\left(\frac{r(x,y)}{\tau}\right)$$

$$= \pi_{\text{gen}}(y|x;\vec{c}) \frac{1}{\mathbb{E}_{\zeta \sim \pi_{\text{gen}}(\cdot|x)}\left[\exp\left(\frac{r(x,\zeta)}{\tau}\right) \frac{\pi_{\text{ref}}(\zeta|x)}{\pi_{\text{gen}}(\zeta|x)}\right]} \frac{\pi_{\text{ref}}(y|x)}{\pi_{\text{gen}}(y|x;\vec{c})} \exp\left(\frac{r(x,y)}{\tau}\right)$$

$$= \pi_{\text{ref}}(y|x) \frac{1}{\sum_{\zeta \in Y} \pi_{\text{gen}}(\zeta|x) \exp\left(\frac{r(x,\zeta)}{\tau}\right) \frac{\pi_{\text{ref}}(\zeta|x)}{\pi_{\text{gen}}(\zeta|x;\vec{c})}} \exp\left(\frac{r(x,y)}{\tau}\right)$$

$$= \pi_{\text{ref}}(y|x) \frac{1}{\sum_{\zeta \in Y} \pi_{\text{ref}}(\zeta|x) \exp\left(\frac{r(x,\zeta)}{\tau}\right)} \exp\left(\frac{r(x,y)}{\tau}\right)$$

$$= \pi_{\text{ref}}(y|x) \frac{1}{\mathbb{E}_{\zeta \sim \pi_{\text{ref}}(\cdot|x)}[\exp(r(x,\zeta)/\tau)]} \exp\left(\frac{r(x,y)}{\tau}\right)$$

$\square$

*Corollary 4.1 Proof.* The proof follows directly from Theorem A.1 and Appendix A.1 in Rafailov et al. [2023]. $\square$

# B Details from Section 5

In this section, we present all omitted details from Section 5.

## B.1 Omitted Proofs from Section 5.1

*Theorem 5.1 Proof.* Let $\vec{r}_i = (r_i(x, y_1), \ldots, r_i(x, y_M))$ be the reward reports of agent $i$ for the $M$ generated candidate sequences. Then, for both allocation rules, holding the candidate sequences and the reports of all other agents fixed, the ex-interim allocation rule (i.e., the probability of returning each of the $M$ generated candidate sequences) from agent $i$'s perspective is:

$$\pi_{\text{int}}(\vec{r}_i; \vec{\beta}_{-i}) = \text{softmax}\left(\frac{\vec{r}_i}{\tau} + \vec{\beta}_{-i}\right), \tag{7}$$

where $\vec{\beta}_{-i,j} = \frac{\sum_{k \in N \setminus \{i\}} r_k(x, y_j)}{\tau} + \log \frac{\pi_{\text{ref}}(y_j|x)}{\pi_{\text{gen}}(y_j|x;\vec{c})}$. Importantly $\vec{\beta}_{-i}$ is an $M$-dimensional vector that does not depend on agent $i$'s reports.

We would like to equip $\pi_{\text{int}}$ with a payment rule $p(\vec{r}_i; \vec{\beta}_{-i})$ so that the resulting interim mechanism will be strategyproof. This requires that $\pi_{\text{int}}$ have a property known as *cyclic monotonicity*. Equivalently, $\pi_{\text{int}}$ must be the (sub)gradient of agent $i$'s utility for bidding truthfully in he mechanism $U(\vec{r}_i; \vec{\beta}_{-i})$, and that utility function must be convex [Frongillo and Kash, 2021, Rochet, 1987, Myerson, 1981].

It is easy to verify that for the function class:

$$U_C(\vec{r}_i; \vec{\beta}_{-i}) = \tau \log \left(\sum_{j=1}^{M} \exp\left(\frac{r_i(x, y_j)}{\tau} + \vec{\beta}_{-i,j}\right)\right) + C, \ C \in \mathbb{R} \tag{8}$$

the allocation rule $\pi_{\text{int}}(\vec{r}_i; \vec{\beta}_{-i})$ is a gradient of $U_C(\vec{r}_i; \vec{\beta}_{-i})$. Additionally, $U_C(\vec{r}_i; \vec{\beta}_{-i})$ is convex in $\vec{r}_i$: The exponential function $e^x$ is (strictly) convex, because its second derivative is positive. The transformation $\frac{r_i(x,y_j)}{\tau} + \vec{\beta}_{-i,j}$ is an affine transformation of $r_i(x, y_j)$, and affine transformations preserve convexity. Finally, it is well-known that the LogSumExp function is convex.

Thus, for any $\vec{\beta}_{-i}$ and for any set of generated candidate sequences, reporting truthfully maximizes agent $i$'s expected utility, with the expectation taken over the draw of the final sequence from the set of candidate sequences. Adopting the quasi-linear utility model, agent $i$'s payment is:

$$U_C(\vec{r}_i; \vec{\beta}_{-i}) = \pi_{\text{int}}(\vec{r}_i; \vec{\beta}_{-i}) \cdot \vec{r}_i - p(\vec{r}_i; \vec{\beta}_{-i})$$

$$p(\vec{r}_i; \vec{\beta}_{-i}) = \pi_{\text{int}}(\vec{r}_i; \vec{\beta}_{-i}) \cdot \vec{r}_i - U_C(\vec{r}_i; \vec{\beta}_{-i})$$

$$p(\vec{r}_i; \vec{\beta}_{-i}) = \pi_{\text{int}}(\vec{r}_i; \vec{\beta}_{-i}) \cdot \vec{r}_i - \tau \log \left(\sum_{j=1}^{M} \exp\left(\frac{r_i(x, y_j)}{\tau} + \vec{\beta}_{-i,j}\right)\right) - C, \ C \in \mathbb{R} \tag{9}$$

$\square$

## B.2 Our mechanism is "almost individually rational"

First, we explain why the standard notion of individual rationality (i.e., weakly positive utility from participation in the mechanism) encountered in most auction settings is impossible to achieve in this domain while converging to the optimal distribution and maintaining incentive compatibility. Then, we explain how, with our payment offset, our mechanism is "almost IR:" In Lemma B.1 we prove that the ex-interim utility of an agent who has zero reward for all candidate sequences and bids truthfully is deterministically zero, i.e., agents that do not contribute to the social welfare (but also do not detract from it) have zero utility. Similarly, in Lemma B.2 we prove that if an agent's reward for all candidate sequences is (weakly) positive, then her ex-interim utility is (weakly) positive.

**Why is indivual rationality (IR) impossible?** *Individual rationality* (IR) stipulates that an agent gains more utility by participating and bidding truthfully in a mechanism than by not participating at all. Typically, if an agent's utility for non-participation is zero, participating should yield weakly positive utility. However, this simplification does not apply in our setting.

As discussed in Section 5.2, agent $i$'s reward for any sequence $y$ can be arbitrarily negative (Equation (10)). The same is true for the utility from truthful participation, as outlined in Equation (8). To ensure a positive utility for every agent in our mechanism, an offset would need to be infinitely large or dependent on agent $i$'s reports. But then the mechanism's allocation rule would no longer be the gradient of agent $i$'s utility with respect to her reports, which would destroy incentive compatibility [Frongillo and Kash, 2021, Rochet, 1987, Myerson, 1981].

It is important to note that this challenge is inherent not just to our mechanism but to any mechanism in this setting that operates with a fixed set of sequences, aims to approximate the optimal distribution, and maintains incentive compatibility. Under these conditions, the only allocation rule that approximates the theoretically optimal distribution (Equation (2)) is that of our mechanism. However, this uniquely determines the agents' utilities, up to a constant factor, as described in Equation (8) [Frongillo and Kash, 2021, Rochet, 1987, Myerson, 1981].

**Lemma B.1.** *For the payment offset $C = -\tau \log \left( \sum_{j=1}^{M} \exp \left( \vec{\beta}_{-i,j} \right) \right)$ if agent $i$'s reward for all candidate sequences is zero, then her ex-interim utility is deterministically zero, for all $\vec{\beta}_{-i} \in \vec{R}_{-i}$.*

*Lemma B.1 Proof.* First, note that for all $\vec{\beta}_{-i} \in \vec{R}_{-i}$, agent $i$'s expected reward for the outcome is zero, as $\pi_{\text{int}}(\vec{r}_i; \vec{\beta}_{-i}) \cdot \vec{r}_i = \pi_{\text{int}}(\vec{r}_i; \vec{\beta}_{-i}) \cdot \vec{0} = 0$. Additionally, agent $i$'s reward for the realized outcome will deterministically be zero, as her reward for all generated candidate sequences is zero. Finally, note that by setting $\vec{r}_i = \vec{0}$ in Equation (9) with the offset $C$ set as in Section 5.3, we have that the agent $i$'s payment is also deterministically zero. Thus, an agent with zero reward for all generated candidate sequences who reports her rewards truthfully has deterministically zero reward for the final outcome and zero payments, and her utility is also deterministically zero. □

**Lemma B.2.** *For the payment offset $C = -\tau \log \left( \sum_{j=1}^{M} \exp \left( \vec{\beta}_{-i,j} \right) \right)$ if agent $i$'s reward for all candidate sequences is positive, then her ex-interim utility is positive, for all reports $\vec{\beta}_{-i} \in \vec{R}_{-i}$.*

*Proof.* Lemma B.1 establishes that when agent $i$'s reward for all candidate sequences is zero, her utility for truthfully bidding in the mechanism, denoted as $U(\vec{0}; \vec{\beta}_{-i})$, is zero for all possible reports of the other agents $\vec{\beta}_{-i} \in \vec{R}_{-i}$.

Furthermore, Theorem 5.1 shows that the mechanism's allocation rule corresponds to the gradient of agent $i$'s utility when bidding truthfully. Because the allocation rule is non-negative, the gradient of agent $i$'s utility for bidding truthfully is also non-negative.

Thus, if agent $i$'s rewards for all candidate sequences are weakly positive, and considering the non-negative gradient of her utility, her ex-interim utility under truthful bidding must be positive, irrespective of the other agents' reports $\vec{\beta}_{-i}$. □

**Corollary B.3.** *For the payment offset $C = -\tau \log \left( \sum_{j=1}^{M} \exp \left( \vec{\beta}_{-i,j} \right) \right)$ if the distribution $\pi_{gen}$ only generates candidate sequences for which agent $i$'s reward is positive, then the ex-ante expected utility of the agent is positive.*

*Proof.* This follows immediately from the fact the the fact that the ex-ante utility of the agent is the expectation of her ex-interim utility with respect to her reward for the generated sequences, and the fact that the second quantity is positive whenever the reward of the agent for all candidate sequences is positive from Lemma B.2. $\qquad\square$

## B.3 Differences from Standard Auction Settings

Standard auction environments typically rely on a set of assumptions that simplify mechanism design; however, these assumptions do not apply to auctions for LLM-generated content. In this section, we detail these assumptions and discuss why they are inapplicable in our context.

First, in a standard auction setting, it is common to assume that the agents' valuation functions satisfy free disposal, i.e., $v_i(S) \geq v_i(S') \; \forall S \supseteq S', S, S' \supseteq \mathcal{I}$. The interpretation of free disposal is that an agent can discard any items she is allocated that she is not interested in. Free disposal combined with the fact that an agent has zero value for the empty bundle mean that her value for any outcome is weakly positive. Second, in most auction environments, the allocation rule is different for different agents: each agent will get allocated her own bundle of items, and we can assume that she is indifferent to the allocation of items to the other agents.

As detailed in Rafailov et al. [2023], assuming that an agent's LLM $\pi_i$ was trained to maximize her reward function (and regularized with respect to its KL divergence from some reference LLM, which we assume to be the same as the auctioneer's reference LLM), there is a one-to-many mapping between an advertiser's optimal LLM, and her implicit reward function. That mapping is:

$$r_i(x, y) = \tau_i \log \frac{\pi_i(y|x)}{\pi_{\text{ref}}(y|x)} + \log Z_i(x) \tag{10}$$

where $Z_i(x)$ is a prompt-dependent constant, and $\tau_i$ is the regularization hyperparameter of agent $i$, similar to the one in Equation (1). All functions in the class defined in Equation (10) are equivalent, in the sense that they induce exactly the same LLM [Rafailov et al., 2023]. This has two implications: First, unlike standard auction environments, an agent's reward can go negative – there is nothing equivalent to the free disposal property. Setting $Z_i(x)$ to zero (which is equivalent to normalizing the induced probabilities by the LLM [Rafailov et al., 2023]), the agent's reward is negative for any sequence for which her LLM assigns a lower probability than $\pi_{\text{ref}}$.

Second, especially in the online advertising application, an agent's expected utility for not participating in the auction is negative: if advertiser $i$ does not participate in the auction, her payment is zero, but her expected value for the outcome is

$$\pi_{\theta_{-i,M}}(\vec{\beta}_{-i}) \cdot \vec{r_i} \tag{11}$$

The other advertisers have very low rewards for the sequences that mention advertiser $i$: assuming their LLMs have been properly trained, they will evaluate all sequences that explicitly mention a different, possibly competing brand, as unlikely. Thus, based on Equation (10) the corresponding advertisers have very low rewards for those sequences and conversely, agent $i$ has low rewards for the sequences that the other advertisers have high rewards for. But based on Equation (2), if advertiser $i$ does not participate in the auction, $\pi_{\theta_{-i,M}}(\vec{\beta}_{-i})$ will assign high probabilities to sequences for which $i$ has low rewards for. Thus, Equation (11) implies that, unlike standard auction environments, the advertiser's expected reward and utility for not participating in the mechanism is negative.

## B.4 "What you give is what you get"

As we explained in Section 5.3.2, Our choice of allocation rule (which is the only allocation rule over a finite set of sequences that converges to the optimal distribution), combined with the fact that the allocation rule is the gradient of the utility to ensure truthfulness, means that agent utilities must also be the same up to potentially agent-specific offsets as indicated by Equation (4):

$$U_C(\vec{r}_i; \vec{\beta}_{-i}) = \tau \log \left( \sum_{j=1}^{M} \exp \left( \frac{1}{\tau} \sum_{k \in N} r_k(x, y_j) \right) + \log \frac{\pi_{\text{ref}}(y_j|x)}{\pi_{\text{gen}}(y_j|x; \vec{c})} \right) + C, \ C \in \mathbb{R} \quad (12)$$

However, not all agents contribute equally to the social welfare of the final outcome. Because of this, implementing the mechanism without a carefully-designed offset would lead to a kind of "reverse market unraveling:" as long as an agent's utility in Equation (12) is positive, she would be incentivized to participate, even if the user query was completely unrelated to her business, because the mechanism would ensure that she received, on expectation, the same (positive) expected utility from doing so as any other participating agent.[6]

Incentivizing unrelated agents to participate would would have adverse effects. First, the better-performing context-aware mechanism would create candidate sequences with worse rewards for *all* agents, because its context would be "diluted" from agents unrelated to the user query. In our running example for the user query "How do I bake cookies?", imagine adding "Try to mention 'EasySwitch', a comprehensive VPN service" to the context of the context-aware LLM $\pi_{\text{gen}}(\cdot|x; \vec{c})$.

Additionally, for both versions of the mechanism, following the discussion in Section 5.2, the agents for whom the user query is unrelated are more likely to have negative rewards for the generated sequences as their LLMs will deem the candidate sequences more unlikely than the reference LLM. Thus, based on Equation (4), their participation in the mechanism will lead to a reduction of the total sum of rewards of the generated sequences for the agents, which will indirectly reduce the expected utility of all agents, making the mechanism less attractive for the user-query-relevant agents.

To summarize, all agents receiving the same utility would incentivize agents for whom the user query is unrelated to participate in the auction. This would in turn reduce everyone's expected utility, potentially reducing the incentive for the user-query-relevant agents to participate, and lead to sequences with worse expected rewards for the agents and usefulness for the user.[7] Thus, in the application of auctions for aggregating agents' preferences over LLM-generated outputs, agents with higher contribution to social welfare also receiving proportionally higher utility by the mechanism is important for the long-term success of the mechanism in practice.

---

[6]We can assume that agents can estimate their expected utility from participation using historical data from past auctions, analogously to how they can estimate their utility for participating in sponsored search auctions.

[7]If we interpret the KL divergence between the distribution induced by the reference LLM and the LLM that generated the candidate sequences as a measure of their expected usefulness for the user.

# C Details from Section 6

## C.1 Detailed Experiment Setup

We create a set of synthetic instances to test our mechanism. Each instance consists of a user query, e.g. "How do I bake cookies?" and a list of advertisers. Each advertiser is defined by an "advertiser name", e.g. "KitchenFix" and an advertiser description, e.g., "producing kitchen appliances."[8] The reference LLM $\pi_{\text{ref}}$ responsible for generating replies that are useful for the agent is Llama-2-7b-chat-hf [Touvron et al., 2023]. The advertisers' LLMs are created using the same reference LLM, and adding the instruction: "Answer the question advertising <advertiser>, <advertiser description>." The context aware LLM is created using the same reference LLM, and adding the instruction: "Answer the query. Try to mention <advertiser 1>, who <advertiser description 1> and <advertiser 2>, who <advertiser description 2>."

Following [Rafailov et al., 2023] the reward function of advertiser $i$ is set to $r_i(x, y) = \log \frac{\pi_i(y|x)}{\pi_{\text{ref}}(y|x)}$, where $\pi_i$ is advertiser $i$'s LLM, i.e., we set $\tau_i = 1, Z_i(x) = 1$ for all advertisers and for all user prompts in Equation (10).[9] For the auctioneer's objective as defined in Equation (1) we set $\tau = 1$, balancing between the advertisers' expected rewards for the generated sequences and the sequences' divergence from the reference LLM responsible for generating useful replies for the user. Thus, the optimal policy according to Equation (2) becomes:

$$\pi_r(y|x) = \frac{1}{Z(x)} \pi_{\text{ref}}(y|x) \exp \left( \sum_{i \in N} r_i(x, y) \right) \tag{13}$$

Following Li et al. [2024], Rozière et al. [2024] we sample from the LLM generating the sequences (either $\pi_{\text{ref}}$ or $\pi_{\text{gen}}$) with temperature $0.8$ and top-p $0.95$. We use $50$ user queries, each with two interested advertisers. To increase the statistical significance of our results, we test each user query on $25$ different random seeds, effectively resulting in $1250$ individual instances. The seed affects both the generation of the candidate sequences, and the randomized sampling of the final sequence from the candidate sequences. We will make all of our code and set of synthetic instances publicly available. Our compute infrastructure is detailed in Appendix C.2.

## C.2 Compute Infrastructure

All experiments were conducted on a compute cluster running Ubuntu 20.04.6 LTS with AMD EPYC processors with 48 cores and 1512GB RAM and Nvidia A100 GPUs and Python 3.12.1.

---

[8]Following Duetting et al. [2023], we use fictitious brand names to avoid the model using any existing ad.
[9]Note that we cannot estimate the true value of $Z_i(x)$. However, it is a query-dependent constant. Given that our experiments focus on the reward difference of different sequences for the same query, this is a non-issue.

## C.3 Comparison of baseline and context-aware mechanism outputs

To provide some intuition of how the context-aware mechanism improves the relevance of generated responses for the advertisers, we present a side-by-side comparison of outputs from both mechanisms for a specific query in Table 1. Note that only the context-aware mechanism is able to incorporate the advertisers into the reply to the user's query.

| **User query** |
|---|
| What are effective ways to learn a musical instrument online? |
| **Advertisers** |
| - MusicMastery: offering online music lessons and virtual tutoring
- InstaTune: selling musical instruments and learning aids |
| **Context-Aware Mechanism Output** |
| Effective ways to learn a musical instrument online include:
1. Online music lessons: Platforms like **MusicMastery** offer one-on-one virtual tutoring, group classes, and pre-recorded lessons. These lessons can be tailored to your skill level and learning style.
2. Interactive learning tools: **InstaTune** offers a range of interactive learning tools, such as apps, software, and games, that make learning fun and engaging.
3. Practice with technology: Utilize digital tools like metronomes, tuners, and recording software to practice and track your progress.
4. Collaborate with other musicians: Join online communities, forums, and social media groups to connect with other musicians and learn from their experiences. . . . |
| **Baseline Mechanism Output** |
| - Practice consistently and regularly: Consistent practice is essential to learning any musical instrument, and it's even more important when learning online. Set aside a specific time each day or week to practice, and make sure to stick to it.
- Find a good teacher or mentor: While it's possible to learn some things on your own, working with a good teacher or mentor can help you improve faster and avoid developing bad habits. Look for someone who is experienced and knowledgeable in the type of music you want to learn.
- Use technology to your advantage: There are many online tools and resources available that can help you learn an instrument. These include video lessons, online tutorials, and apps that can help you practice and track your progress. . . . |

Table 1: Comparison of context-aware and baseline mechanism outputs using 10 candidate sequences. Advertisers highlighted in bold, outputs truncated to conserve space.

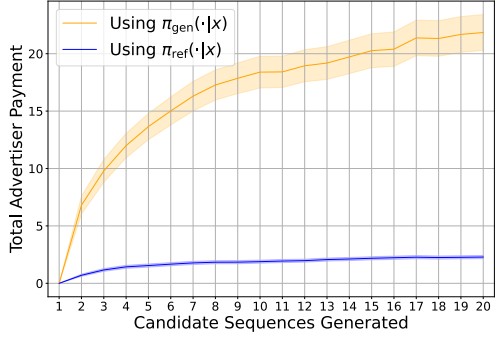
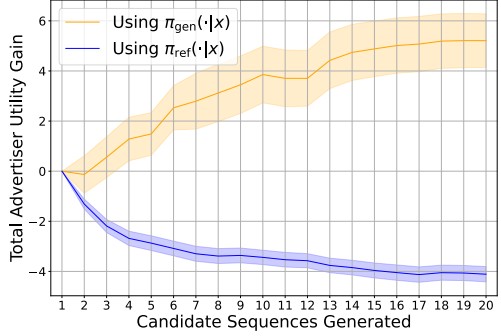

(a) Total advertiser payments as a function of the number of candidate sequences.

(b) Total advertiser utility gain from participation as a function of the number of candidate sequences.

Figure 2: Total advertiser payments and utility gain from participation as a function of the number of candidate sequences generated using $\pi_{\text{ref}}$ and $\pi_{\text{gen}}$. Averaged over 1250 runs including 95% CIs.

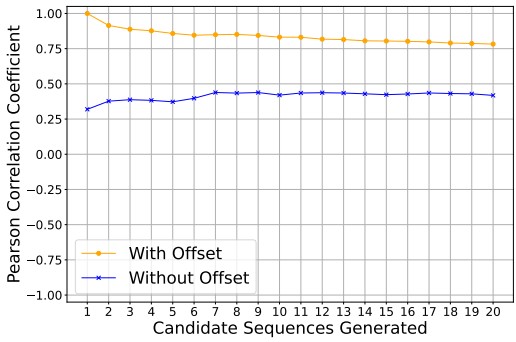
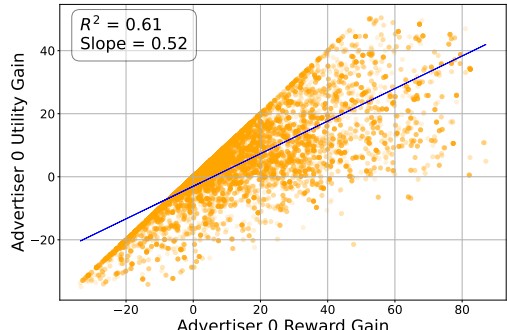

(a) Pearson correlation between advertiser utility and reward gain from participation as a function of the number of candidate sequences.

(b) Scatter plot of advertiser reward and utility gain from participation. We additionally show a linear regressor fit to that data, its slope and its $R^2$.

Figure 3: Analysis of the joint distribution of advertiser utility and reward gain from participation using the context-aware LLM $\pi_{\text{gen}}$ to generate the candidate sequences.

### C.4 Evaluating the Effectiveness of the Payment Rule

In this section, we examine the impact of our payment rule introduced in Section 5. We focus on how the offset of Section 5.3 aligns each advertiser's utility with her contribution to the social welfare.

Figure 2 presents the empirical evaluation of our payment rule that incorporates the agent-specific offset of Section 5.3. In Figure 2a we plot the auctioneer's revenue (i.e., the total payment by the advertisers) as a function of the number of generated candidate sequences, for both the context-aware and baseline versions of our mechanism. In Figure 2a, we observe that this payment rule, for our context-aware mechanism, results in significant revenue for the auctioneer, which increases rapidly with the number of generated candidate sequences. Furthermore, if we compare the advertisers' total payment in Figure 2a with their utility gain (i.e., reward gain minus payment) from participation in Figure 2b, we see that our payment rule successfully converts a significant portion of the surplus created for the advertisers into revenue for the auctioneer. At the same time, Figure 2b illustrates that our context-aware mechanism, coupled with this payment rule, results in positive utility gain from participation in the auction for the advertisers: the rewards gained exceed their payments. Finally, the advertisers' utility gain from participation also increases with the number of candidate sequences.

Figure 3 explores the effectiveness of the payment offset introduced in Section 5.3 in aligning the advertisers' contributions to social welfare with their utility gains. In Figure 3a we plot the Pearson

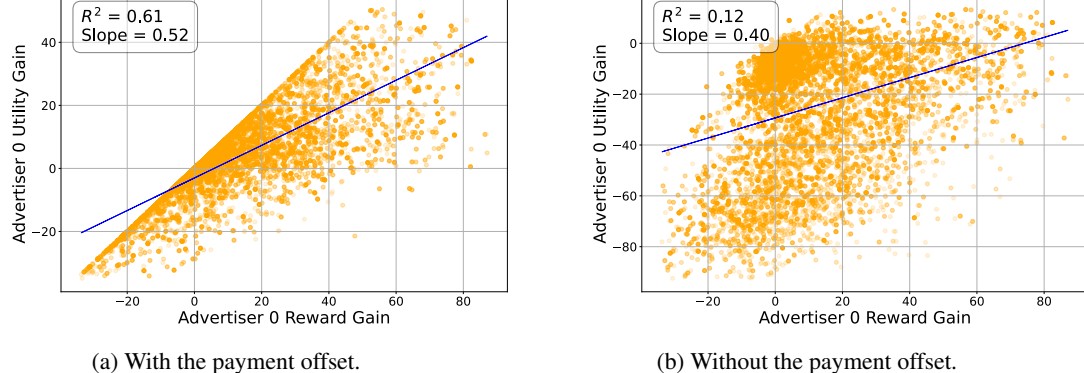

(a) With the payment offset.

(b) Without the payment offset.

Figure 4: Comparative scatter plots of advertiser reward and utility gain from participation, with and without the payment offset of Section 5.3 for candidate sequences generated by the context-aware LLM $\pi_{\text{gem}}$. We additionally show a linear regressor fit to that data, its slope and its $R^2$.

correlation between an advertiser's reward and utility gain from participation in the mechanism. We compare the Pearson correlation of these two metrics for the incentive compatible payment derived in Theorem 5.1 with and without the agent-specific offset defined in Section 5.3, as a function of the number of candidate sequences generated in our context-aware mechanism. In Figure 3a we observe that, for all numbers of candidate sequences, the agent-specific offset results in a significant increase in the correlation of an agent's reward and utility gain from participation. Additionally, we observe that for all numbers of candidate sequences, the Pearson correlation between the agent's reward and utility gain, using the agent-specific offset, remains above $0.75$. This is a clear indication that, with the agent-specific offset, there is a very strong linear correlation between the two metrics. This strong linear correlation indicates a more equitable mechanism, as higher contributions to social welfare directly translate to greater utility gains for advertisers.

Further substantiating this, Figure 3b features a scatter plot of advertiser rewards versus utility gains for all tested problem instances, as well as a linear regressor fitted to that data, its slope and coefficient of determination.[10] The regressor's large coefficient of determination of $0.61$ indicates that the regressor is able to fit the datapoints well, suggesting that, for our context-aware mechanism with the payment offset, the relationship between advertiser utility and reward gain is quite linear. Additionally, the positive slope of the regressor indicates that the correlation is positive. In Appendix C.5 we provide a comprehensive comparison of the relationship between an advertiser's reward and utility gain from participation, with and without the offset, for both the context-aware and baseline versions of our mechanism. In all cases, the introduction of the agent-specific payment offset introduced in Section 5.3 makes the relationship between an advertiser's reward and utility gain from participation significantly more linear and positively correlated.

### C.5 Comprehensive Experimental Evaluation of the Offset from Section 5.3

In Figure 4 we compare the scatter plots of the advertiser reward and utility gain from participation in the mechanism, with and without the payment offset introduced in Section 5.3 for candidate sequences generated using the context-aware LLM $\pi_{\text{ref}}$. Additionally, for both subfigures, we show a linear regressor fitted to the data, as well as its slope and coefficient of determination. Comparing the two subfigures, it is immediately obvious that adding the offset to the payments makes the relationship between advertiser reward and utility gain far more linear. This is confirmed by the coefficient of determination of the linear regressors fit to each dataset. The coefficient of determination of the linear regressor is far larger when we use the offset. Without the payment offset, the coefficient of determination is almost $0$, indicating that, without our payment offset, reward gain is not a predictive measure of an agent's utility. Additionally, the slope of the linear regressor is also higher for the scatter plot with the payment offset. In Figure 5 we make the same comparison, but for candidate

---

[10]In all cases we plot all instances where more than $4$ candidate sequences were generated. Furthermore, we exclude the top and bottom $0.5$ percentile of both metrics to avoid extreme outliers.

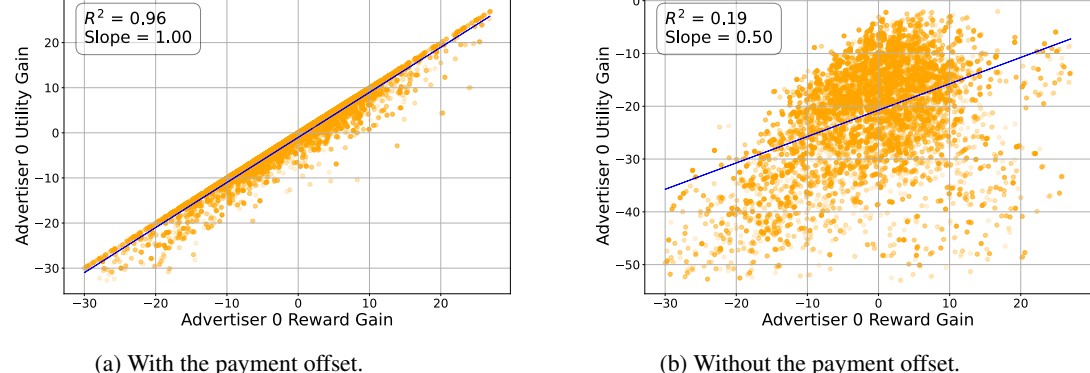

(a) With the payment offset.

(b) Without the payment offset.

Figure 5: Comparative scatter plots of advertiser reward and utility gain from participation, with and without the payment offset of Section 5.3 for candidate sequences generated by the reference $\pi_{\text{ref}}$. We additionally show a linear regressor fit to that data, its slope and its $R^2$.

sequences generated using the reference LLM $\pi_{\text{ref}}$. The results are now even more pronounced. In Figure 5a we observe the relationship between advertiser utility and reward gain with our payment offset is almost perfectly linear, as suggested by the linear regressor fitted to the data having a slope of $1.00$ and an extremely high coefficient of determination of $0.96$, indicating that it can almost perfectly fit the data. Without our payment offset however, in Figure 5b we can see that the relationship between the two metrics is again both less linear, and less positively correlated, as the slope of the linear regressor is $0.5$ and its coefficient of determination is only $0.19$. To conclude, in all cases tested, the use of the agent-specific offset introduced in Section 5.3 makes the relationship between an advertiser's contribution to welfare and her allocation both more linear and more positively correlated.

