# OpenReview forum: "Truthful Aggregation of LLMs\\ with an Application to Online Advertising"
_ICML.cc/2024/Workshop/Agentic_Markets — Agentic Markets @ ICML'24 Poster_

### Official Review · Reviewer_MESn · 2024-06-13
**Truthful Aggregation of LLMs with an Application to Online Advertising**

**Rating:** 7
**Confidence:** 4

**Review:**

# Summary
This paper studies a setting in which an auctioneer must supply an randomised answer to a user’s input prompt. Informally, the goal of the auctioneer is to select an answer which is close in distribution to a trusted reference LLM, whilst also modifying the answer to account for a set of advertisers who hold preferences over the set of possible answers. The formal objective of the auctioneer closely resembles loss functions used in reinforcement learning from human feedback and captures this trade-off. The auctioneer has the ability to demand payments from each advertiser based on their selected answer and the preferences reported. Summarising, the auctioneer is tasked with devising an allocation rule which selects a good distribution of answers given the reports of each advertiser and an effective payment rule which ensures that each advertiser reports their preferences truthfully.
Practically speaking, the auctioneer cannot hope to return the optimal distribution over answers as this problem is intractable given that the space of prompts is extremely large. Instead, the authors propose a sample-based allocation rule in which a second LLM is used to generate a finite set of $M$ candidate answers. The auctioneer then computes the optimal distribution for the resulting empirical version of the auction in which one of the $M$ candidates must be returned as the answer. When $M$ is taken to infinity, the authors demonstrate that their allocation rule converges to the optimal distribution for the original auction.
In addition, using a cyclic monotonicity argument, the authors devise a family of accompanying payment rules that guarantee truthfulness. Moreover, the authors then describe how a specific payment rule can be selected from this family in order to satisfy nice properties such as (almost) individual rationality. Lastly, the authors empirically demonstrate the performance of their mechanism in the domain of online advertising.

# Strengths
The online advertising setting provides an excellent motivation for the theoretical contributions of the paper. Moreover, the authors do a good job of discussing existing work and outlining how their approach offers several significant advantages (especially wrt Duetting et al). The fact that the proposed mechanism does not need access to LLM model weights hugely increases the utility of the work.
The mechanism is simple, but elegantly designed, leaning on ideas from RLHF and traditional mechanism design. Moreover, it smartly leverages the fact that LLMs are fast to sample from but intractable to marginalise over, by solving an empirical version of the original auction.
Overall, I think the paper delivers a neat solution to a practical problem which will soon become an inevitable reality in online advertising.

# Weaknesses
Whilst the paper combines ideas from disparate fields, the technical methods employed to derive theoretical results are very standard (though this is a minor grievance).
This is more of a question than a criticism. The authors mention that VCG is inapplicable in this setting as the optimal distribution cannot be tractable identified. However, couldn’t a similar approach to the authors’ be used to transform the VCG mechanism? That is instead of running VCG over the original auction, you first sample a subset of answers. Then, you run VCG on this new empirical auction. Wouldn’t the resulting auction be strategyproof in this case? Note that I am not advocating for this, as I assume VCG soon becomes intractable as $M$ grows, but it might be worth thinking about.
It is rather disappointing that the authors made no effort to adapt their paper to the 4 page recommendation made by the conference organisers. Likewise, it is disappointing that the authors are using the NeurIPS template rather than the ICML one. I strongly recommend the authors switch their formatting over to the ICML template when submitting a final version.
No guarantees are provided for the auctioneer’s revenue. The authors provide several guarantees that show the auction mechanism is attractive for advertisers, but at the end of the day it is the auctioneer who decides upon the mechanism that is used. I think it would be interesting to start thinking about the revenue guarantees your auction provides.

# Other Comments / Suggestions
The setting of the authors assumes that advertisers have a reward function describing the desirability of each answer. This assumption seems quite strong as humans often find it difficult to assign numerical values to outcomes. It may be worth thinking about a variation of this setting where agent preferences are ordinal and not cardinal.
The subheading of Section 5.1 mentions convexity, but the following paragraph does not mention convexity at all. As a result, it is unclear to readers who have not read the appendix why convexity is even being mentioned. I would recommend briefly mentioning that you use a cyclic monotonicity argument so that the relevance of convexity is clear to new readers.
Name your mechanism. It will make it easier to refer to. For example, instead of using “Our Mechanism” in each subheading you can start using the mechanism’s name.
I think there should be some way to characterise the finite sample performance of your mechanism in terms of how well the context-aware LLM matches the true optimal distribution. I say this because the problem looks in many ways similar to importance sampling wherein you generate samples for one distribution by sampling from another one

---

### Official Review · Reviewer_MpsV · 2024-06-18
**1. #20: Truthful Aggregation of LLMs with an Application to Online Advertising:**

**Rating:** 8
**Confidence:** 4

**Review:**

This paper presents an advancement to a novel approach in combining bidding for LLMs by multiple advertiser 'agents' and proves that under a certain marketplace setting, it is incentive-compatible to participate in the bidding marketplace for any strategy by the advertiser. The authors propose a rigorously grounded theory and a small set of experiments to validate their proposed model for fair auctions over conditionally generated output from an LLM.

To me, it is hard to follow where the authors discuss whether a particular candidate sequence is close to the set of sequences in the absence of advertisements. This would help showcase that under their proposed bidding mechanism, advertisements can be limited from becoming 'clickbait' which is a fundamental problem with advertising today, highly optimized for keywords and ostensibly organic product placement. Maybe I missed a plot?

While I have not evaluated the lemmas and proofs, the discussion is clear, problem formulation makes sense to me and is an adaptation of RLHF for finetuning LLMs. Modulo the usage of some subjective terms like 'hopeless' and 'nice properties' which the authors should address (I only have a minor quibble with them), the paper provides a materially useful contribution that can inform the development of bidding marketplaces for LLMs, which is quite an important topic from a real-world utility standpoint.

I recommend accepting it as an Oral Talk (4/5), with medium confidence. The community can benefit from their discussion.